

# The distribution and isotopomeric characterization of nitrous oxide in the Eastern Gotland Basin (central Baltic Sea)

Pratirupa Bardhan[1,2], Claudia Frey[3], Gregor Rehder[4] and Hermann W. Bange[1]

[1]GEOMAR Helmholtz Centre for Ocean Research Kiel, Wischhofstr.1-3, 24148 Kiel, Germany
[2]now at Dept of Geology and Environmental Science, University of Pittsburgh, 4107 O' Hara St, PA 15260, USA
[3]Department of Environmental Sciences, University of Basel, Bernoullistr. 30, 4056 Basel, Switzerland
[4]Leibniz Institute for Baltic Sea Research Warnemünde (IOW), Seestr. 15, 18119 Rostock, Germany

*Correspondence to*: Pratirupa Bardhan (pratirupabardhan@gmail.com)

**Abstract.** Nitrous oxide (N$_2$O) is a greenhouse gas with a global warming potential ~300 times that of carbon dioxide.

Coastal areas are important sources of N$_2$O to the atmosphere but the biogeochemical pathways of N$_2$O production and

consumption are not well understood. We measured the concentrations and nitrogen (N) and oxygen (O) stable isotopes

(d$^{15}$N and d$^{18}$O) of N$_2$O in the Baltic Sea to constrain the sources and sinks of N$_2$O in this system. Further, we used the

intramolecular $^{15}$N variation or the site preference (SP) as additional tracer. Samples were taken at 7 stations during a cruise

with R/V Elisabeth Mann Borgese to the Eastern Gotland Basin (central Baltic Sea) in May/June 2019. The isotope

signatures of N$_2$O accumulation in the oxycline reflected production predominantly via ammonia oxidation. In the waters

where hydrogen sulfide (H$_2$S) was detected, active N$_2$O consumption was implied by pronounced decrease in N$_2$O levels in

tandem with enrichments in $\delta^{15}$N$_{bulk}$, $\delta^{18}$O and SP. High site preference values of N$_2$O observed in suboxic waters of the

stations where H$_2$S buildup was minimal to non-detectable point to the possibility of non-canonical denitrification pathways

mediated by fungi or abiotically. A sedimentary source of N$_2$O was observed in those stations, which resulted in a

decoupling of the $\delta^{15}$N$_{bulk}$ and $\delta^{18}$O of N$_2$O. Our results reveal that transient oxygenation events have the potential to modify

the N cycling within the oxic-anoxic interface even if for shorter time scales.

## 1 Introduction

Nitrous oxide (N$_2$O) is an important climate-relevant atmospheric trace gas: in the troposphere it acts as a greenhouse gas

(IPCC, 2021) and in the stratosphere it is one of the major precursors for ozone depletion (Ravishankara et al., 2009).

Nitrous oxide has a global warming potential (GWP) which is ~300 times larger than that of carbon dioxide ($CO_2$) over a 100-year time scale (IPCC, 2021). Atmospheric $N_2O$ mole fractions have risen in the past 100 years due to increased anthropogenic influence (Ravishankara et al., 2009; Flückiger et al., 1999).

The ocean is a major (~20%) natural source of $N_2O$, albeit poorly characterized (Tian et al., 2024; Yang et al., 2020). Within the marine environment, coastal seas, including estuaries, are considered important as sources of atmospheric $N_2O$ and play a major role in its global budget (Resplandy et al., 2024, Rosentreter et al., 2023). Thus, it is crucial to improve our knowledge and understanding of these systems. However, existing literature on the magnitude, distribution, seasonality and environmental controls of $N_2O$ production from these systems is still limited.

In the open and coastal oceans, $N_2O$ is produced via various pathways: In oxygenated waters, $N_2O$ is formed as a byproduct during nitrification (i.e. the stepwise microbial ammonia oxidation to nitrate) (Nevison et al., 2003; Yoshinari, 1976). The positive correlation between oversaturation of dissolved $N_2O$ (expressed as $\Delta N_2O$ and representing the excess $N_2O$ relative to the concentration in equilibrium with the ambient atmosphere) and apparent oxygen utilization (AOU) is often used as indirect evidence of $N_2O$ production via nitrification in oxic waters (Yoshinari, 1976, Nevison et al., 2003). The largest oceanic $N_2O$ concentrations and atmospheric fluxes were found in coastal upwelling regions associated with the oxygen deficit zones (ODZs) of the Indian, Eastern Tropical North Pacific and Eastern Tropical South Pacific Oceans (Naqvi et al., 2000; Arévalo-Martínez et al., 2015; Suntharalingam and Sarmiento, 2000; Nevison et al., 1995). In these systems, denitrification, the stepwise microbial reduction of nitrate to dinitrogen gas ($N_2$), produces $N_2O$ as an intermediate (Cohen and Gordon, 1979; Ward et al., 2009). During suboxic conditions, $N_2O$ is reduced to $N_2$ in the last step of denitrification thus acting as a sink for $N_2O$ (Körner and Zumft, 1989). Under oxygen-deficient (i.e. suboxic or sulfidic) conditions, the linear relationship of $\Delta N_2O$: AOU, therefore, breaks down due to enhanced $N_2O$ yield by both nitrifiers (Lipschultz et al., 1981) and denitrifiers (Knowles et al., 1981) as well as consumption of $N_2O$ by denitrifiers. Thus, it is a challenging task to distinguish the pathways of $N_2O$ production in low-$O_2$ waters where nitrifying and denitrifying microbes can co-exist (Ji et al., 2015).

The stable nitrogen and oxygen isotopes signatures of $N_2O$ (expressed as $\delta^{15}N$ and $\delta^{18}O$ respectively) serve as effective

natural tracers for identifying the sources and sinks of $N_2O$, because its isotopic composition provides valuable insights in at

least three ways: (i) The bulk isotopic composition: The isotopic makeup of the initial substrate influences the bulk isotopic

composition of $N_2O$. For example, during ammonia oxidation by nitrifiers, the $\delta^{15}N$ and $\delta^{18}O$ of $N_2O$ are determined by the

$\delta^{15}N$ of the source ammonium ($NH_4^+$) and the $\delta^{18}O$ of dissolved $O_2$, respectively. In the case of nitrifier-denitrification

(microbial ammonia oxidation to nitrite followed by stepwise reduction to $N_2$) and denitrification, the $\delta^{15}N$ and $\delta^{18}O$ of $N_2O$

are influenced by the isotopic signature of the source nitrate ($NO_3^-$) or nitrite ($NO_2^-$). (ii) The kinetic isotope effect (e): The

process of isotopic fractionation i.e. where lighter isotopes ($^{14}N$ and $^{16}O$) are preferentially taken up during product

formation, resulting in the substrate becoming enriched in the heavier isotopes ($^{15}N$ and $^{18}O$) which also affects the stable

isotopic composition of $N_2O$. Laboratory and field data report a wide range of values for N and O isotope effects during the

production and consumption of $N_2O$ (Lewicka-Szczebak et al., 2015). (iii) The site-specific nitrogen isotopic signature: $N_2O$

has a linear and asymmetrical structure ($N_\beta =N_\alpha$ -O) and the difference in $\delta^{15}N$ values of the central ($N_\alpha$) and outer ($N_\beta$)

positions is referred to as site preference (SP). Unlike the bulk $\delta^{15}N$ and $\delta^{18}O$ of $N_2O$, SP is independent of the source

substrate and is determined solely by the process involved (Frame and Casciotti, 2010). As a result, $N_2O$ produced through

nitrifier-denitrification and denitrification exhibits low SP signatures (-11 to 0 ‰) while $N_2O$ generated from ammonia

oxidation has high SP signatures (30 to 36 ‰).

The Baltic Sea waters can serve as a natural laboratory to study the biogeochemistry of $N_2O$ using a stable isotope approach.

The first study on $N_2O$ concentrations from the Baltic Sea was conducted in the Western Gotland Basin (Rönner, 1983), and

1500 nM $N_2O$ was observed when the bottom water at one station turned anoxic (Rönner, 1983). This is one of the highest

reported concentrations until today (for comparison: $N_2O$ equilibrium concentrations usually range from 5 to 15 nM).

Another study (Walter et al., 2006), extensively covering the southern and central Baltic Sea, reported buildup of $N_2O$ when

the system became oxygenated after a prolonged sulfidic period. The authors attributed the onset of nitrification to cause this

$N_2O$ buildup in the water column. More recently, this was confirmed after the last major inflow into the Baltic Sea in 2015

(Myllykangas et al., 2017). Short-term buildup of very high (> 500 nM) $N_2O$ concentrations was observed immediately

before the bottom waters lost dissolved $O_2$ again. Enhanced $N_2O$ production (Walter et al., 2006) has been observed during

the transition from suboxic to oxic conditions, which, when coupled with a simultaneous buildup of hydroxylamine

(Schweiger et al., 2007), led to the conclusion that nitrification, specifically ammonia oxidation, is the predominant $N_2O$

source. Long term monitoring (Ma et al., 2019) at the Boknis Eck Time-Series Station (Eckernförde Bay, SW Baltic Sea) has

also revealed the seasonality of $N_2O$ concentrations with high concentrations in winter and early spring and lower

concentrations during the suboxic/sulfidic periods in autumn. Thus, the variability of $N_2O$ in the Baltic Sea is spatially and

temporally complex. A first, albeit concise, data set of isotopic and isotopomeric ratios of $N_2O$ and $N_2O$ production by

ammonia oxidation at Boknis Eck was presented in a method article by Ji and Grundle (2019).

The specific questions that we address here are: (1) What are the dominant pathways of $N_2O$ production and consumption in

the oxic-anoxic transition zone of the Baltic Sea water column? (2) How effective is the stable isotopic composition of $N_2O$,

including site preference, as a tool to distinguish between the processes involved?

## 2 Study Site and Methods

**2.1 Study site and sample collection**

The Baltic Sea consists of several interconnected basins that vary widely in the extent of oxygen deficiency (Meier et al.,

2017). The Gotland Basin is the largest basin with a maximum depth of 240 m. Due to limited water exchange and strong

thermohaline water column stratification, the central and southern parts of the Baltic Sea are typically suboxic (Liblik et al.,

2018) and even sulfidic (with high levels of hydrogen sulfide, $H_2S$). Occasionally, the North Sea waters flow in over the sills

and flush the deeper basins. These inflow events are known as MBIs (major Baltic inflow), and they bring oxygen-rich and

saline waters to the deeper basins of the southern and central Baltic Sea. In the recent past, the MBIs have been occurring

roughly once in a decade (Gräwe et al., 2015) although this statement has been questioned by Mohrholz (2018) who found a

decadal variability of MBIs with a timescale of 25-30 years. The most recent MBI before our sampling campaign, which was

also the third largest one in 60 years, occurred in December 2014 (Liblik et al., 2018; Dellwig et al., 2021). In addition,

weaker inflows of saline waters can lead to intrusions in intermediate water depths of the major basins. Freshwater input

occurs as well through large river runoff and the combined input of saline North Sea waters and the riverine freshwater

renders the Baltic Sea to be a brackish water system, one of the largest of its kind (Weckström et al., 2017). The Baltic Sea is

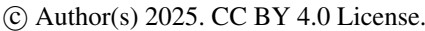


also vulnerable to eutrophication, and the oxygen deficiency in the deeper basins has intensified not only in volume and frequency but also in magnitude by spreading to the coastal areas (Voss et al., 2011; Meier et al., 2019). In 2019, the year of our sampling campaign, the area of the suboxic zone in the Baltic Sea of >80,000 km$^2$ was one of the three largest on record (Hansson et al., 2019).

Samples were collected onboard R/V Elisabeth Mann Borgese from May 20[th] to June 5[th], 2019 (Cruise EMB214) as part of the Baltic Sea project EU BONUS INTEGRAL. For this study, six stations were sampled along a transect in the Eastern Gotland Basin (Fig. 1). Station 25 is the deepest at 233 m, followed by Station 30 at 98 m depths. The remaining four stations (26, 27, 28 and 29) have depths ranging from 80 to 90 m. The basin is permanently stratified with the halocline extending from 50 to 100 m. The transect and sampling was specifically selected to cover the oxic-anoxic transition zone at high resolution, and to comprise stations where this transition zone interacts with the sediment, an area which is characterized by enhanced microbial turnover processes (Noffke et al., 2016). The seventh station is Station 32, outside the Gotland Basin, where the halocline (40 - 70 m) was quite steep and bottom waters were more saline (15-17) than the bottom waters of the other stations (11 - 13). This station was in the Bornholm Basin south of the Eastern Gotland Basin and chosen as reference station without H$_2$S accumulation to understand the spatial changes in N$_2$O isotopomer biogeochemistry within the Baltic Sea.




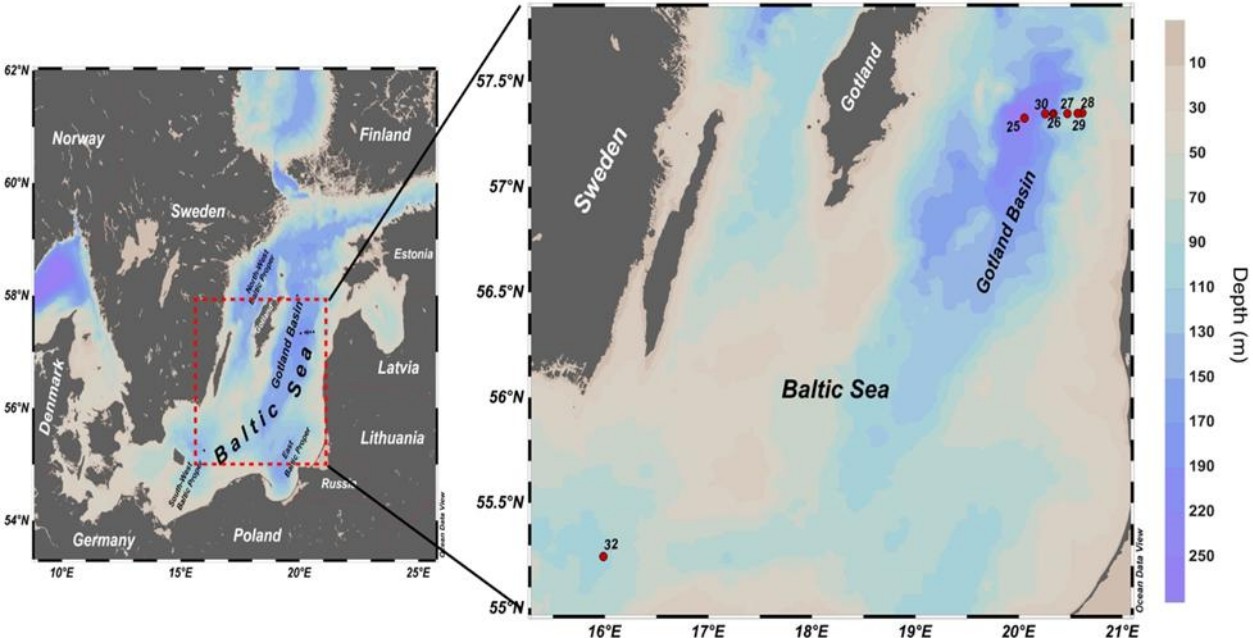

**Figure 1: Study sites in the Eastern Gotland Basin, central Baltic Sea, sampled during cruise 214 on RV Elisabeth Mann Borgese**
**in May/June 2019.**

Water samples were collected in special 5L Free Flow water bottles, developed by IOW/HYDROBIOS for sampling in systems with strong vertical gradients, mounted on a rosette equipped with double sensor packages for conductivity, temperature and pressure (CTD) and oxygen sensors. Oxygen was analyzed by Winkler titration on enough samples to

assure proper calibration of the oxygen sensors. The CTD SBE 43 oxygen sensors recorded oxygen concentrations that were validated frequently by Winkler titration results. Dissolved nutrients, including $NO_3^-$ and $NO_2^-$, were measured onboard from filtered samples using standard photometric methods by means of an autoanalyser (Grasshoff et al. 1999). $H_2S$ was determined spectrophotometrically by the methylene blue reaction (Grasshoff et al., 1999).

Samples for dissolved $N_2O$ were taken in 125 mL glass septum vials with overflow and closed with gray butyl stoppers and

aluminium crimps avoiding the introduction of bubbles. Samples were then treated with 100 µL saturated mercuric chloride solution to inhibit microbial activity until analysis. All $N_2O$ concentration data were directly measured on board within 36 hours after sampling.



Single samples for dissolved N$_2$O isotopes were collected into 160 mL glass serum bottles. A Tygon® tubing was attached to the Niskin bottle, and the serum bottles were filled and allowed to overflow twice taking care not to introduce bubbles.

Samples were poisoned with 100 µL saturated mercury chloride (HgCl$_2$) solution and then capped with gray butyl stoppers and aluminium crimps. They were shaken well and stored in the dark at 4°C until analyses.

## 2.2 Dissolved N$_2$O concentrations and atmospheric mole fractions

The dissolved N$_2$O concentrations were determined using a dynamic headspace method, i.e. a purge and trap system linked to a gas chromatograph to allow for the simultaneous measurement of N$_2$O and CH$_4$. In brief, approximately 10 mL of the

samples were transferred into a purge vessel using a calibrated air-tight syringe without contact to air (volume error <0.5%). The dissolved gases were stripped out of the sub-sample using an ultrapure helium purge stream, and cryo-focused. Through heating, the trapped gases were injected onto the gas chromatographic system, the N$_2$O was isolated and measured on an electron capture detector. The method is described in detail in Wilson et al. (2018) and Sabbaghzadeh et al. (2021). The estimated precision was determined to be better than 2% for N$_2$O (Sabbaghzadeh et al., 2021).

The N$_2$O saturations (%) were calculated as

$$N_2O_{sat} = 100*N_2O_{observed}/N_2O_{equilibrium}$$

where the N$_2$O$_{equilibrium}$ is the equilibrium concentration of N$_2$O calculated according to Weiss and Price (1980) with the in-situ temperature, salinity and the mean monthly atmospheric mole fraction of N$_2$O (332.9 ppb) for May and June 2019. The atmospheric mole fractions of N$_2$O at the time of the sampling were taken from the NOAA/ESRL monitoring station in Mace

Head (Ireland) (http://www.esrl.noaa.gov/gmd/).

## 2.3 Stable isotope methods

Bulk N$_2$O isotope and isotopomer analyses were conducted at the Department of Environmental Sciences, University of Basel, Basel, Switzerland. Using helium (He) as carrier gas, N$_2$O was purged from the sample vials into a customized purge-and-trap system (McIlvin and Casciotti, 2011) and analyzed by continuous-flow IRMS (GC-IRMS, Thermo Delta V). Ratios

of m/z 45/44, 46/44, and 31/30 were converted to $\delta^{15}$N-N$_2$O (referenced to air), $\delta^{18}$O-N$_2$O (referenced to Vienna Standard Mean Ocean Water, VSMOW), and site-specific $\delta^{15}$N$^\alpha$ and $\delta^{15}$N$^\beta$-N$_2$O (Frame and Casciotti, 2010; Mohn et al. 2014; Kelly



et al. 2023) using three isotopic mixtures of N$_2$O in synthetic air (CA06261: $\delta^{15}$N = −35.74 ‰, $\delta^{15}$N$^\alpha$ = −22.21 ‰, $\delta^{15}$N$^\beta$ = −49.28 ‰, $\delta^{18}$O = 26.94 ‰; Fl.53504: $\delta^{15}$N = 48.09 ‰, $\delta^{15}$N$^\alpha$ = 1.71 ‰, $\delta^{15}$N$^\beta$ = 94.44 ‰, $\delta^{18}$O = 36.10 ‰; and CA08214: $\delta^{15}$N = 6.84 ‰, $\delta^{15}$N$^\alpha$ = 17.11 ‰, $\delta^{15}$N$^\beta$ = −3.43 ‰, $\delta^{18}$O = 35.39 ‰; kindly provided by J. Mohn, EMPA, Switzerland).

Standard deviations for triplicate measurements of our standards were ± 0.39 ‰ for $\delta^{15}$N$_{bulk}$-N$_2$O, ± 0.56 ‰ for $\delta^{18}$O-N$_2$O and ± 1.29 % for SP-N$_2$O.

## 3 Results

The surface layer (0-50 m) of the Eastern Gotland Basin was well oxygenated with O$_2$ concentrations (>300 µM) being near equilibrium with the atmosphere (Fig. 2). The oxycline extended from 50 to 70 m in most stations (up to 75 m in Stations 28

and 29). Below the oxycline, the waters gradually turned suboxic ([O$_2$] < 20 µM). It is important to mention that at Station 25, we observed a second smaller layer of oxygenated water ([O$_2$] = 29 µM) in a depth of 120 m. H$_2$S concentrations did not exceed 0.5 µM at Stations 26, 27, 29 and 30 and were not detected at Stations 28 and 32. Station 25, which was also the deepest station, had the highest H$_2$S concentration (4.7 µM) in the bottom waters. Based on a definition of the suboxic zone of [O$_2$] < 20 µM (Paulmier and Ruiz-Pino, 2009), its thickness varied from only 4 m (Station 28) up to > 100 m (Station 25)

(Fig 2).





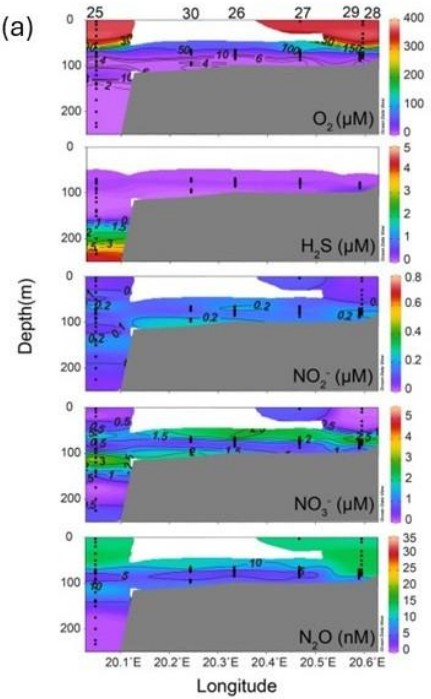
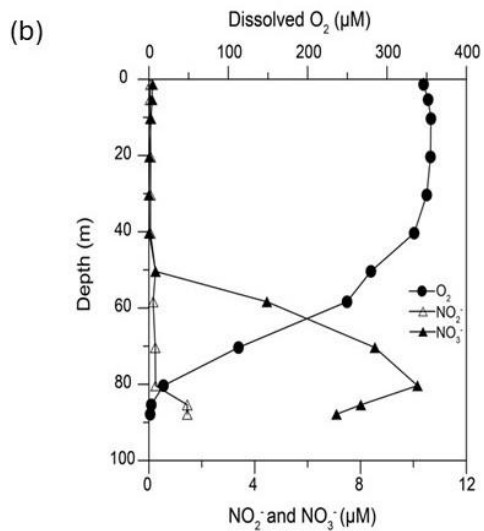

**Figure 2: Hydrographic transects of oxygen, nitrate, nitrite and hydrogen sulfide at Stations (a) 25, 30, 26, 27, 29, 28 and (b) profiles of these parameters at Station 32. Hydrogen sulfide was not detected at Station 32 and hence not depicted in Fig. 2b.**


The surface waters were depleted in nitrate and nitrite with the highest concentrations being 0.84 µM and 0.12 µM respectively (Fig. 2). The nitrate maxima were observed at 70-75 m and highest nitrate concentrations ranged from 3.5-8.5 µM. Nitrate consumption was observed below the nitrate maxima. The bottom waters of Stations 25, 29, 28, 26, and 27 had nitrate concentrations below 1 µM. At Station 30, the nitrate levels dropped in the suboxic zone ($O_2$ between 9-2 µM) before

increasing up to ~3 µM in the bottom depths ($O_2 < 3$ µM). At Station 25 a second nitrate peak coincided with the $O_2$ intrusion at 120 m. Nitrite moderately increased (0.5-1 µM) in the oxycline for all stations. At Station 32, the nitrate concentrations were higher (~10 µM) at the nitrate maximum compared to other stations and at the time of sampling, the bottom depths had high concentrations of nitrate (7 µM) and nitrite (1.5 µM). An overlap of $H_2S$ and $NO_3^-$ was present in 5





out of the 7 stations (25, 26, 27, 29, 30). In general, no distinct secondary nitrite maximum (SNM) was detected at all

stations, similar to observations by Frey et al. (2014a).

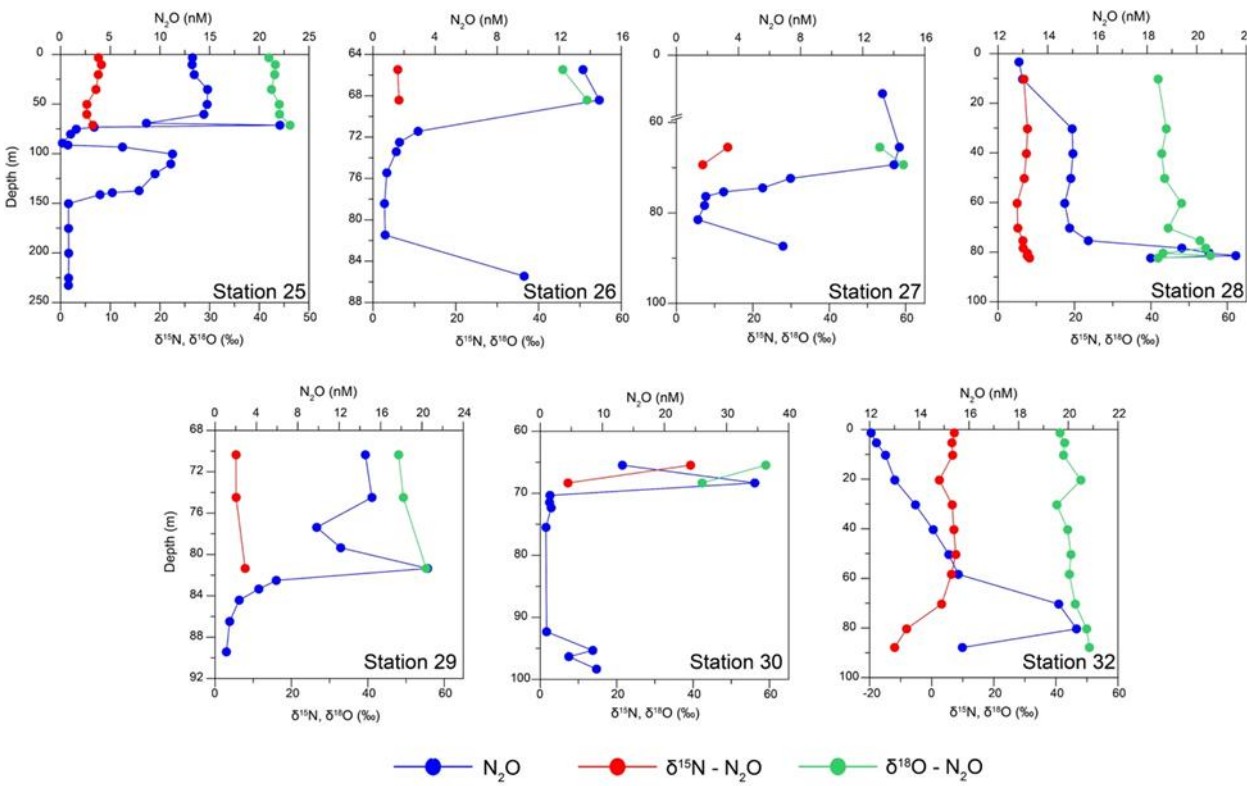

**Figure 3: Depth profiles of N₂O concentrations (in blue) and its δ¹⁵N$_{bulk}$- (red) and δ¹⁸O (green) isotopes at Stations 25, 26, 27, 28, 29 30 and 32.**

Surface water N₂O concentrations ranged between 10-15 nM (Fig. 3). These waters were almost at atmospheric equilibrium

(94-104 % saturation, Table S1). In the oxycline the N₂O concentrations increased to 15-20 nM at the top of the ODZ. The

N₂O saturation remained in an almost similar range as surface water (98-105 % with respect to atmospheric N₂O). Beyond

the oxycline, at some stations (26, 27, 29 and 30), the N₂O concentrations steadily declined to <1 nM (N₂O saturation <10%

with respect to atmospheric N₂O). An increase in N₂O concentrations was recorded at the bottom depths at Stations 26, 27,

and 30. At the deepest station, Station 25, the N₂O concentration profiles demonstrated a second peak, coinciding with the

intrusion of oxygenated water, and then decreased to <1 nM in the bottom depths. In the near-bottom waters of Stations 28 and 32 the $N_2O$ concentrations were in the range of 16-22 nM ($N_2O$ saturation 125-150 % with respect to atmospheric $N_2O$).

The mean $\delta^{15}N_{bulk}$ (6.6 ± 1.8 ‰) and $\delta^{18}O$ (43.1 ± 2.1 ‰) of $N_2O$ in surface waters were close to tropospheric $N_2O$ values (~

6.6 ‰ and 44.2 ‰, Toyoda et al., 2013) (Fig 3). The former remained nearly the same (6.6 ± 1.9‰) in the oxycline as the $N_2O$ concentrations increased while the latter increased to 46.5± 4.6 ‰. Below the oxycline, the $\delta^{15}N_{bulk}$-$N_2O$ moderately increased to 7.1 ± 0.9‰ accompanied by a decrease in $N_2O$. The mean $\delta^{18}O$-$N_2O$ also increased up to 49.6 ± 5.1 ‰ in the ODZ waters. At Station 32, extremely depleted $\delta^{15}N_{bulk}$ up to -12 ‰ were recorded in the bottom waters. The average values of $\delta^{15}N_{bulk}$ and $\delta^{18}O$ at the maximum $N_2O$ concentration were 7.1 ± 0.6 ‰ and 51.5 ± 5.9 ‰ respectively.

In the surface waters, the mean SP was 18.1 ± 9.3 ‰ (Table S1). Like the $\delta^{18}O$-$N_2O$, SP increased to 30.2 ± 7.5 ‰ in the oxycline (Table S1). Below the oxycline, the SP displayed a lot of variability. The SP values displayed maxima in the suboxic waters in general. The exception was Station 25 where the values dropped to less than 0 ‰. The mean $\delta^{15}N^{\alpha}$ was 14.8 ± 5.4‰ in the surface waters. It increased to 18.8 ± 4.9‰ in the oxycline. In the bottom waters, the $\delta^{15}N^{\alpha}$ increased to 30-50 ‰ with a few low values recorded at Station 25. The mean $\delta^{15}N^{\beta}$ values in the surface and the oxycline waters were -

3.3 ± 4.6 ‰ and -6.7 ± 4.1 ‰ respectively. The values further decreased to -10 to -20 ‰ in the ODZ waters with the lowest value of -32.5 ‰ recorded in the bottom depths of Station 32 coincident with highly depleted $\delta^{15}N_{bulk}$-$N_2O$.

In the suboxic waters of this transect in the Baltic Sea, the $N_2O$ profiles generally depicted a rapid decline concurrent with declining dissolved oxygen concentrations. This presented a methodological challenge as these low concentrations (~1 nM $N_2O$) were below the threshold for reliable isotopic measurements. In the limited set of datapoints that we could measure,

these are the main trends that appeared: 1) A moderate enrichment in $\delta^{15}N_{bulk}$- $N_2O$ in all the stations except Station 32, with declining $N_2O$ concentrations. 2) A decoupling between $\delta^{18}O$ and $\delta^{15}N_{bulk}$- values at Stations 28 and 32. 3) A peak in $N_2O$ concentrations in the bottom waters at Stations 26, 27, 30 and 32. 4) Highly depleted $\delta^{15}N_{bulk}$- values in suboxic depths at Station 32. We will address each trend and discuss these results in the following section.



# 4 Discussion

## 4.1 N₂O in oxic waters


Surface N₂O saturations in the Eastern Gotland Basin ranged from 92 to 104 % with a mean of 98.8 ± 3.7 % in the month of June 2019 which showed that surface waters were near equilibrium with the atmosphere and thus did not represent a source or sink of N₂O to the atmosphere. The production of N₂O through nitrification, along with decreasing dissolved oxygen concentrations, was indicated by increasing $NO_3^-$ and N₂O concentrations beneath the surface waters (between 65-70 m for

Stations 26 and 27; and between 50-75 m for Stations 25, 28, 29 and 32). There was no significant linear relationship of $DN_2O$ and AOU in oxic waters which implies that nitrification rates were low and counterbalanced by the air-sea exchange of N₂O (Fig 4). The low $DN_2O$ at high AOU values were comparable to those typically found in not nitrifying suboxic or sulfidic waters of the Baltic Sea (see e.g. Walter et al., 2004) (Fig. 4).

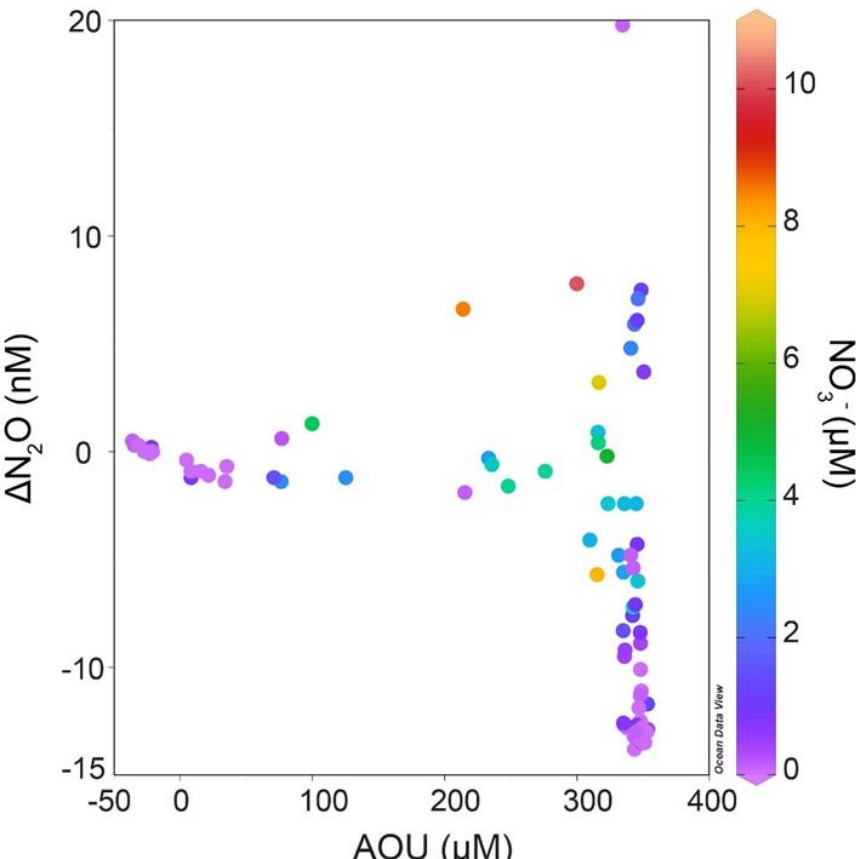



**Figure 4: ΔN₂O/AOU relationship from all the stations color-coded with dissolved nitrate concentrations.**

The potential source of $N_2O$ in oxic waters can be determined from the intercepts of the linear regression between the inverse of the observed $N_2O$ concentration ($1/N_2O_{observed}$) and the $\delta^{15}N_{bulk}$, $\delta^{18}O$ or SP observations (Keeling, 1961; Fujii et al., 2013). We applied this approach known as the Keeling plot method to the surface (0-50m) and the oxycline waters (50-70m) (dissolved $O_2$ concentrations > 20 µM in all samples), but no significant linear trend was visible for the three isotopes of $N_2O$

(Fig 5). Most data points scatter around the isotopic composition of $N_2O$ in air rendering this as a dominant source in oxic waters. Note, that this method cannot be applied in suboxic waters, where consumption of $N_2O$ is dominating. The $\delta^{15}N_{bulk}$ of $N_2O_{produced}$ were higher in the surface waters (9.7 ‰) and closer to the atmospheric equilibrated value than in the oxycline (5.3 ‰). If nitrification is a source of $N_2O$, then the $\delta^{15}N_{bulk}$-$N_2O_{produced}$ should be lower to and similar to the $\delta^{15}N$ of the $NH_4^+$ substrate. Frey et al. (2014a) reported $\delta^{15}N$-$NH_4^+$ values in the range of 6-10 ‰ in the upper suboxic zone and up to 22

‰ at the redoxcline in the Gotland Basin. The kinetic isotope effect of ammonia oxidation to nitrite, the first step of nitrification, is $^{15}\varepsilon_{NH4+} = 14\text{-}38\%$ (Casciotti et al., 2003). Considering the $\delta^{15}N$-$NH_4^+$ of 22 ‰ (in the redoxcline) from Frey et al. (2014a) and the mean $\delta^{15}N_{bulk}$ of $N_2O_{produced}$ of 5.3 ‰), the kinetic isotope effect in this dataset falls in the range of 8-33‰ making nitrification a likely source.



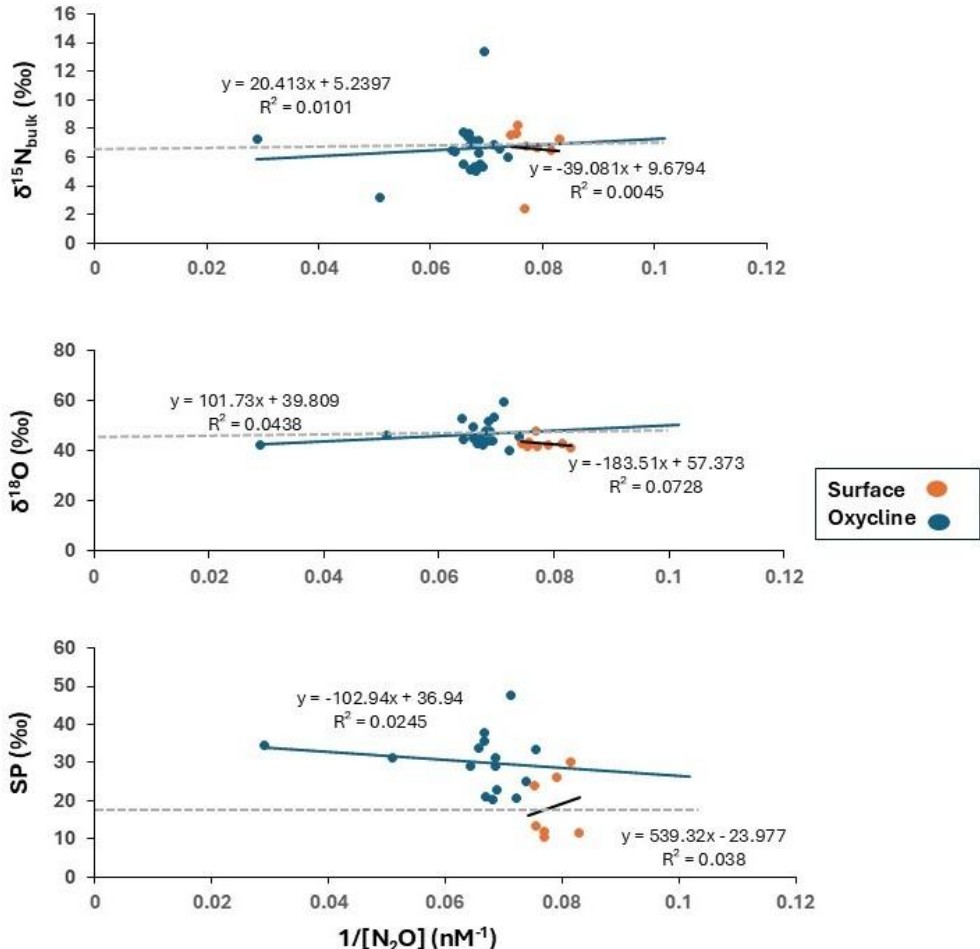

**Figure 5: Linear regressions of $\delta^{15}N_{bulk}$, $\delta^{18}O$ and SP against 1/(N$_2$O concentration). Regressions were performed on two groups of data: surface (0-50m) (represented by orange circles) and the oxycline (50-70m) (represented by blue circles). Tropospheric N$_2$O has been represented as a grey dashed line with values reported by Toyoda et al. (2013) ($\delta^{15}N_{bulk}$ = ~ 6.6‰, $\delta^{18}O$ = ~44‰, and SP = ~18‰).**

The $\delta^{18}O$ of N$_2$O$_{produced}$ was lower (39.8‰) in the oxycline than in the surface (57.4 ‰). The N$_2$O molecule derives its oxygen from dissolved O$_2$ and H$_2$O molecules during nitrification and from nitrite or nitrate during denitrification (Ostrom et al., 2000). Moreover, the $\delta^{18}O$-N$_2$O is also impacted by the isotopic fractionation during N$_2$O production and O isotope equilibration (Frame and Casciotti, 2010; Casciotti and Buchwald, 2012). The $\delta^{18}O$-NO$_x$ and the $\delta^{18}O$-H$_2$O in the central



Baltic Sea were reported as ~0.1 ± 1.8 ‰ and -6 ± 0.4 ‰ respectively (Frey et al., 2014a), so $\delta^{18}O$ of $N_2O_{produced}$ are higher than potential sources and are indicating a depleted $^{18}O$ source during nitrification.

We observed an increase of SP from the surface to the oxycline. SP is process-dependent and substrate-independent. SP during production via nitrification is usually in the range 30-38 ‰ and during production via denitrification and nitrifier-denitrification is in the range -10–25 ‰ (Sutka et al., 2004; Frame and Casciotti, 2010). The SP of $N_2O_{produced}$ (Fig. 5) increased from 23.9 to 36.9 ‰ from the surface to the oxycline. The mean SP of $N_2O$ in the atmosphere is 18.7±2.5‰ (Toyoda et al., 2013), suggesting its predominance in surface waters. However, as depth increases, the observed rise in SP

appears to be linked to the production by nitrification, because the mean SP values in the oxycline waters were closer to the SP values for ammonia oxidation as compared to nitrifier-denitrification. However, based solely on SP it is difficult to draw conclusions whether ammonia-oxidizing archaea (AOA, SP ~30 ‰) or ammonia-oxidizing bacteria (AOB, SP ~36 ‰; Santoro et al., 2011; Sutka et al., 2003) are dominating. Nonetheless, based on previous studies, which have found high-level expression of archaeal nitrification genes (Thaumarchaeota, related to the genus *Nitrosopumilus*) in the Baltic Sea above the

redoxcline (Labrenz et al., 2007) as well as high activities (Berg et al., 2015), AOA may be potential contributors to $N_2O$ production. The AOA are probably more dominant due to their ability to cope with frequent exposure to sulfidic waters (Berg et al., 2015; Jäntii et al. 2018) as compared to the AOB, which are more prevalent in the nutrient rich coastal waters (Happel et al., 2018).

To sum it up, atmospheric $N_2O$ is an important source in the oxic waters. The increase in $N_2O$ concentrations in the

subsurface waters along with decline of $O_2$ concentrations and increase in the $NO_3^-$ concentrations implies in-situ $N_2O$ production by bacterial or archaeal ammonia oxidation as indicated by the $\delta^{15}N_{bulk}$-, the $\delta^{18}O$ and the SP of $N_2O$. Our results align with those of Ji and Grundle (2019), who observed an increased yield of $N_2O$ due to increasing ammonia oxidation under decreasing $O_2$ concentrations. The authors reported the highest rate of $N_2O$ production coincided with the lowest in-situ $O_2$ concentration. The nitrifier-denitrification pathway seems to be of minor significance in this zone. The isotopic

compositions were also quite similar between the surface and the oxycline which renders the possibility of exchange between these layers with a potential for supersaturation and high surface flux of $N_2O$.



## 4.2 N$_2$O in suboxic waters

Microbial denitrification proceeds by the stepwise reduction of $NO_3^-$ to $NO_2^-$ to NO to N$_2$O and ultimately to N$_2$. Thus, denitrification acts as both a source and sink for N$_2$O. Chemolithoautotrophic and heterotrophic denitrification are the two dominant processes of fixed nitrogen (N) removal in the Baltic Sea redoxcline (Frey et al., 2014a; Hannig et al., 2006; Bonaglia et al., 2016; Dalsgaard et al., 2013). When H$_2$S and $NO_3^-$ coexist in this zone, fixed N removal is fueled through the chemolithoautotrophic mode. Heterotrophic denitrification can be the dominant mode of fixed N removal in the Baltic Sea especially when the sediment slope is steep, which increases the occurrence of internal waves (Bonaglia et al., 2016). H$_2$S concentrations were quite low as compared to some of the studies conducted during the stagnant periods (Frey et al., 2014a). A few inflows were recorded in 2019 including one in June reaching the Eastern Gotland Basin (SMHI, 2020), which may have caused lower H$_2$S accumulation. The recent intrusion of a layer of oxygenated water with its core at ~110m water depth is visible in our transect. Dalsgaard et al. (2013) performed a set of incubation experiments and observed N$_2$O to be increasing during denitrification with increasing amounts of sulfide. In our study of natural samples, however, we did not find such a correlation because H$_2$S concentrations were below 0.5 µM when co-existing with $NO_3^-$. However, both modes of denitrification can be incomplete and stop at N$_2$O, whether one has higher N$_2$O yields is not known. Additionally, the isotope fractionation effect on N$_2$O production during incomplete chemoorganotroph and chemolithotrophic denitrifiaction or N$_2$O consumption during complete denitrification must be considered. The N and O isotopic effect for N$_2$O produced during canonical denitrification using nitrate or nitrite as substrate are 10 - 39‰ and -40 - -4‰ respectively (Casciotti et al., 2002; Toyoda et al., 2005; Sutka et al.,2006). The negative O isotope effect is due to the preference of the produced N$_2$O to retain the $^{18}$O within the N$_2$O bond and release the $^{16}$O instead. The N$_2$O, when reduced to N$_2$, causes an enrichment in $\delta^{15}N_{bulk}$- and $\delta^{18}$O-N$_2$O values as well as an increase in SP signatures respectively (Ostrom et al., 2007; Yamagashi et al., 2007).

For ease of discussion, we can roughly divide the stations into two groups: at Stations 28 and 32, no detectable sulfide could be measured (Group A) and at Stations 25, 26, 27, 29 and 30, sulfide was detected and co-existent with nitrate below the oxycline (Group B). While there was variability in isotopomeric signatures within these stations, a common feature of the

former group was the accumulation of $N_2O$ observed in the bottom waters. In the latter group, rapid consumption of $N_2O$ limited its isotopic measurements.

### 4.2.1 Group A: Stations with no detectable sulfide

Stations 32 and 28 comprise Group A. Station 32 is located outside the Eastern Gotland Basin (in the Bornholm Basin) and

has greater proximity to the North Sea. It is possible that smaller inflows (SMHI, 2020) may have ventilated the deep water at this station. In general, anoxic conditions in the Bornholm Basin are known to be seasonal in nature and not as persistent as in the central Baltic Sea. A decoupling of the $\delta^{15}N_{bulk}$- and $\delta^{18}O$-$N_2O$ was observed in the bottom waters of Station 32. In the suboxic bottom waters (80-88 m), the $^{15}N_{bulk}$ became more depleted and the $^{18}O$ became more enriched with decreasing $N_2O$ concentrations (Fig. 7). These depths also recorded a pronounced buildup of nitrite (1.5 µM, Fig. 2b) that was not

observed in the other stations.

To explain the depleted $\delta^{15}N_{bulk}$- values in Station 32, we look at the precursors of $N_2O$. The $\delta^{15}N$ of nitrate, the presumed precursor to $N_2O$, were 8-10 ‰ (Supplementary data, Fig S2) at Station 32 and do not explain the unusually low values. The $\delta^{15}N$ of ammonium, another possible precursor, was reported to be between 5 and 10 ‰ (Frey et al., 2014a). The dual isotope signatures of dissolved nitrate exhibited progressive enrichment concomitant with nitrate consumption which points

to occurrence of denitrification (Fig. S2, Supplementary Information).

As the consumption of $N_2O$ during denitrification involves breakage of only the $N_\alpha$-O bond, the $\delta^{15}N_\alpha$ and the $\delta^{18}O$ should increase while $\delta^{15}N_\beta$ should remain unchanged. In these waters, however, while $\delta^{15}N_\alpha$ exhibited a moderate increase, the $\delta^{15}N_\beta$ was observed to decrease (Fig. 6). The SP showed a positive correlation with the $\delta^{18}O$-$N_2O$ (with $R^2 = 0.97$) (deepest 3 data points in Fig 6b) which suggests that the process that led to enrichment of $\delta^{18}O$ also caused a depletion of $\delta^{15}N_\beta$.

Moreover, the correlations between $\delta^{18}O$-$N_2O$ and the $\delta^{15}N_{bulk}$-, $\delta^{15}N_\alpha$ and $\delta^{15}N_\beta$ were all negative thus suggesting co-occurrence of multiple processes at this station: one that consumes $N_2O$ rendering the $\delta^{18}O$ more enriched while another introduces $^{15}N$ depleted nitrogen into $N_2O$. Toyoda et al. (2005) have observed differential isotopic fractionation of N incorporation into α and β positions in a particular strain of a denitrifying bacterium. Similar observations of declining $\delta^{15}N_\beta$ and increasing $\delta^{18}O$ and SP have been reported from the sulfidic waters of the Black Sea (Westley et al., 2006), the eastern




tropical North Pacific Ocean (Yamagishi et al., 2007) and the coastal surface waters of the monsoonal upwelling region of

the Arabian Sea (Naqvi et al., 1998; 2006) and have been attributed to shifts from N₂O consumption to net production.

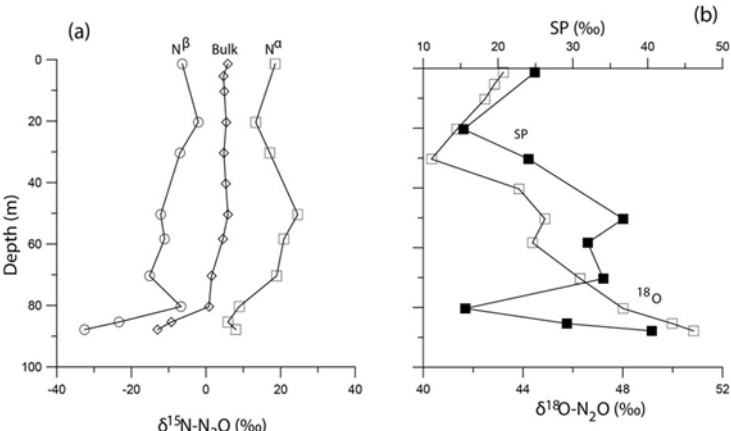

**Figure 6: The isotopomeric composition of N2O at Station 32. Panel (a) shows the depth profiles of δ¹⁵Nbulk-, δ¹⁵Nα and δ¹⁵Nβ, panel (b) shows the depth profile of δ¹⁸O and SP**

While the $\delta^{15}N_{bulk}$- and $\delta^{18}O$-N₂O were significantly positively correlated with the $\delta^{15}N$- and $\delta^{18}O$-NO₃⁻ respectively, the slopes were significantly lower than 1, thus implying the co-occurrence of multiple processes. A close and immediate coupling of nitrification and denitrification in these waters was already suggested by Frey et al. (2014a).

At Station 28, N₂O accumulated in the suboxic waters. While we do observe a decreasing trend of dissolved nitrate with depth, which could explain the production of N₂O, no N₂O consumption was observed. As the enzyme N₂O reductase,

responsible for reducing N₂O to N₂, is highly sensitive and may be inhibited by even nanomolar O₂ concentrations (Dalsgaard et al., 2014), incomplete denitrification could cause an accumulation of N₂O in these depths. A decoupling of the $\delta^{15}N_{bulk}$- and $\delta^{18}O$-N₂O values was observed at this station as well. While the enrichment in $\delta^{15}N_{bulk}$- and $\delta^{15}N_{\alpha}$ values indicate N₂O consumption, the depletion of $\delta^{18}O$ and $\delta^{15}N_{\beta}$ values points towards production of N₂O.



The SP values of the N$_2$O in the suboxic depths of the Group A stations were in the range 26-40‰. Unlike bulk N$_2$O

isotopes, the SP values are independent of the precursor molecules. Fungal denitrification and iron-mediated

chemodenitrification are noncanonical N$_2$O production pathways that have unique SP signatures as compared to

heterotrophic denitrification (SP = -11 to 0‰, Frame and Casciotti, 2010). The SP values of fungal denitrification and

chemodenitrification have been reported to be in the range of 20-37‰ (Rohe et al., 2014) and 10-22‰ (Grabb et al., 2017)

respectively. Fungal denitrification typically ends at N$_2$O due to the missing N$_2$O reductase (Nos) enzyme in most fungi

(Shoun et al., 2012) and could explain the accumulation of N$_2$O observed in the bottom waters. The higher SP values in this

pathway are due to the enzyme involved in the reduction of NO to N$_2$O, the P450NOR. In case of chemodenitrification,

coastal and estuarine sediments are favourable hotspots because of their dynamic redox fluctuations due to the presence of

active iron cycles (Wankel et al.,2017). 15–25% of the total N$_2$O production in the marine sediments from a coastal area of

the Baltic Sea called the Norsminde Fjord in Denmark has been attributed to this process (Otte et al., 2019).

Also, in case of Stations 28 and 32, since these observations were recorded in the bottom waters, benthic N$_2$O production

may also play a significant role. The bottom waters were suboxic which means sedimentary nitrification and/or

denitrification were possible N$_2$O sources. Previous studies in the Eastern Gotland Basin (Hylén et al., 2022; Myllykangas et

al., 2017) observed sedimentary efflux of N$_2$O, which was attributed to incomplete denitrification. The authors observed that

the large intrusion of oxygenated water during 2015 and several small inflows in the following years resulted in aeration of

the previously long-term sulfidic sediments of the Eastern Gotland Basin. Further, algal aggregates were found to be

hotspots for seafloor N$_2$O production (Hylén et al., 2022).  While the reason behind the spatial variability of N$_2$O buildup in

the bottom waters is not clear, it is evident that the microbial processes at the sediment-water interface should be considered

in budget models for more accurate output.

### 4.2.2 Group B: Stations with detectable sulfide

As mentioned earlier, at Stations 25, 26, 27, 29 and 30, it was a challenge to characterize N$_2$O isotopomerically due to

rapidly declining concentrations, which we were not able to capture with our depth resolution. However, for the measured

N$_2$O, we observed increasing $\delta^{15}N_{bulk}$- and $\delta^{18}O$ values concomitant with reduction in nitrate in the low-oxygen waters of

these stations, indicating consumption of N$_2$O via denitrification (heterotrophic/chemolithoautotrophic or both) as observed



elsewhere (Farías et al., 2009; Casciotti et al., 2018). During denitrification, when $N_2O$ gets reduced to $N_2$, the O-$N_\alpha$ bond

breaks and both $\delta^{15}N_\alpha$- and $\delta^{18}O$-$N_2O$ are expected to increase with an expected slope of 1.7-1.9 in their linear equation

(Ostrom et al., 2007) while the bond-breakage is expected to have little effect on $\delta^{15}N_\beta$. As a result, the SP is expected to

increase too. However, in our dataset there was a negative trend of $\delta^{15}N_\alpha$- vs. $\delta^{18}O$-$N_2O$, the slope was 1 and correlation was

significant (Fig. S1, Supplementary information) which may be explained by diffusion-induced $^{15}N$ depletion in $N_2O$ prior to

reduction (Lewicka-Szczebak et al., 2014) and/or $N_2O$ reduction and production occurring in close proximity within the

same microsite (Ostrom et al., 2007). $\delta^{15}N_{bulk}$ ($\varepsilon^{15}$) and $\delta^{18}O$ ($\varepsilon^{18}$) were determined by performing linear regressions vs. -

ln[$N_2O$] assuming a closed system Rayleigh model (Fig. 7). The $\varepsilon^{15}$ and $\varepsilon^{18}$ were -4.48 ‰ (r = 0.36, p> 0.1) and -9.34 ‰ (r=

0.11, p>0.1) respectively. The reported $\varepsilon^{15}$ and $\varepsilon^{18}$ values for $N_2O$ consumption are 4-13 ‰ and 11-31 ‰ (Barford et al.,

1999; Ostrom et al., 2007; Yamagishi et al., 2007).

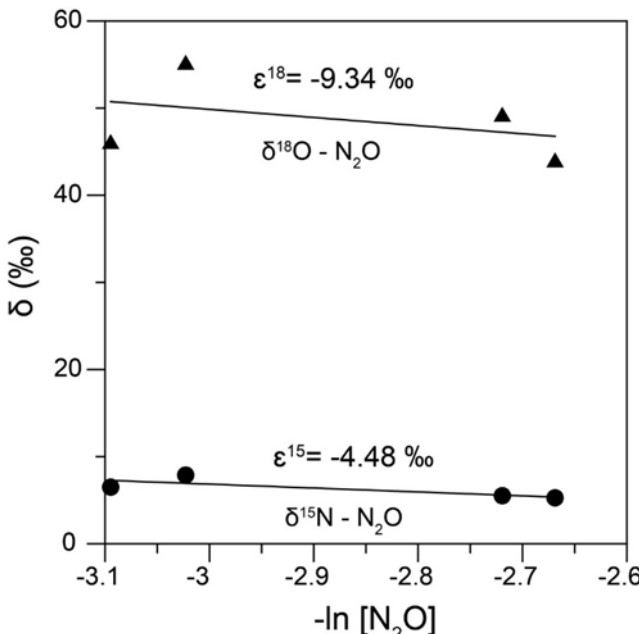

**Figure 7: The N and O isotope effects during $N_2O$ consumption at Stations 25,26,27,29 and 30 obtained by plotting the $\delta^{15}N_{bulk}$- (filled circles) and $\delta^{18}O$ (filled triangles) against -ln[$N_2O$].**

The $\varepsilon^{18}/\varepsilon^{15}$ ratio of $N_2O$ reduction during microbial denitrification has been observed to be ~2.5 in a wide range of aquatic systems and irrespective of the metabolic mode (lithotrophic vs. heterotrophic) and a value $\varepsilon^{18}/\varepsilon^{15} = \sim 2.1$ in our study is an indication of $N_2O$ reduction to be predominant at these stations (Wenk et al., 2016).

Moreover, the low isotope effect values could be an intrinsic feature of the Baltic Sea redoxcline due to diffusion limitation, which has been found for $NO_3^-$ isotopes along the redoxcline previously (Frey et al. 2014a). In a culture study on a chemolithoautroph belonging to a group of the *Epsilonproteobacteria*, considered to be the major denitrifiers in the Baltic Sea redoxcline (Bruckner et al., 2013), the lower apparent N:O isotopic enrichment factor in nitrate was proposed to be caused the periplasmic nitrate reductase enzyme Nap (Frey et al., 2014b). The enzyme responsible for $N_2O$ reduction is known as $N_2O$ reductase (NosZ Clade I) which is also located in the bacterial periplasm like the Nap. This implies that diffusion limitation is a potential factor. Data on the $N_2O$ isotope systematics of marine chemolithoautotrophic denitrifiers are limited with only one published report available to the best of our knowledge (Li et al., 2024). In this study the authors reported a distinct SP signature (~5.1‰) of the chemoautotrophic denitrification from a series of enrichment experiments from freshwater lakes.

Additional factors that can also impact $N_2O$ isotopic signatures are activity of nosZ II genes, and other pathways. Microbes hosting the NosZ II Clade genes, known as $N_2O$ reducers, cannot produce $N_2O$ due to a lack of other denitrifying enzymes like nitrate and nitrite reductases, but they possess the NosZ Clade II enzyme to reduce $N_2O$ to $N_2$ (Jones et al., 2013). Although previously reported in several soil-based studies, the Clade II genes were found to be more abundant than the Clade I types in the suboxic Chesapeake Bay waters (Tang et al., 2022), the Pearl River estuary (Hu et al., 2023) and the ODZ of the eastern tropical South Pacific Ocean (Sun et al., 2017). A comprehensive study on the abundance of NosZ Clade II in the Baltic Sea is currently unavailable. Additionally, the isotope effects for NosZ clade II are unknown. We have already mentioned fungal and chemodenitrification and their unique SP signatures during $N_2O$ production in the previous section. Dissimilatory nitrate reduction to ammonium (DNRA) may be another $N_2O$ source to be considered (Streminska et al., 2012). Bonaglia et al. (2016) found evidence of DNRA at the Eastern Gotland Basin redoxcline. The isotope effects and SP values of $N_2O$ produced via DNRA have recently been characterised by Xu et al. (2024).

## 5 Concluding remarks

Mitigating $N_2O$ emissions will depend on identifying microbial pathways of $N_2O$ production and their constraints. Sporadic intrusions of $O_2$-enriched water masses into the deep basins of the central Baltic Sea bring about distinct transformations in the water column nitrogen cycling and the underlying processes. Isotopic tracer profiles of $N_2O$ provided insight into its

origin and cycling in the Baltic Sea waters. Production of $N_2O$ occurred in the oxycline via nitrification (ammonia oxidation). Simultaneous production and consumption of $N_2O$ in the suboxic zone and bottom waters could be attributed to benthic incomplete denitrification. The isotope signature in $N_2O$ identified active $N_2O$ reduction but could not differentiate between chemolithoautotrophic and organotrophic denitrifiers. Our results demonstrated the spatial variability of the N-loss processes within our study area in the Baltic Sea. While this study provided some answers, it also raised several questions

and directions for future research. Culture experiments of Baltic Sea chemolithoautotrophs to investigate their $N_2O$ isotope systematics will be a crucial next step. Further investigations on the impact of transient oxygenation events on the pelagic N loss should also be executed. We observed $N_2O$ production at the sediment-water interface in this area where the depth of the pelagic redoxcline is close to the sediment surface (i.e. coincides with the water depth). Future research should consider other biotic (e.g. fungal denitrification) and abiotic (e.g. chemodenitrification, chemical hydroxylamine oxidation) $N_2O$

formation processes. The results may be implemented in global and regional biogeochemical models to understand the response of $N_2O$ production and consumption pathways to various environmental stressors (e.g. eutrophication and deoxygenation).

*Data availability.* All data will be made available upon request.

*Author contributions.*PB, GR and HWB designed the study. GR was the principal investigator during Cruise EMB214. GR performed the sample collection and data curation of water chemistry parameters including $N_2O$ concentrations. PB and CF performed the analysis of the nitrous oxide isotopomers. CF performed isotopomeric data correction. PB, GR and HWB contributed to the funding. PB wrote the manuscript and all authors contributed to the writing, review and editing.



*Competing interests.* At least one of the (co-)authors is a member of the editorial board of Biogeosciences. The authors have no other competing interests to declare.

*Acknowledgements.* This project was funded by the EU BONUS INTEGRAL project. It received funding from BONUS (Art 185), funded jointly by the EU, the German Federal Ministry of Education and Research, the Swedish Research Council Formas, the Academy of Finland, the Polish National Centre for Research and Development, and the Estonian Research Council. We thank the captain, the chief scientist and the crew onboard the R/V EMB214 cruise for their professional assistance at sea. Special thanks to Lars Kreuzer (IOW) for the nutrient, oxygen and $H_2S$ analyses, and to Stefan Otto and Sarah Velasco-Sobeck (IOW) for the $N_2O$ concentration measurements performed at sea. We are very grateful to Moritz Lehmann at the University of Basel for providing the opportunity to use his laboratories. Thomas Kuhn at University of Basel and Thomas Hansen at GEOMAR are gratefully acknowledged for their technical support. PB thanks the Alexander von Humboldt Foundation for providing a postdoctoral fellowship (grant number 1204748).

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
