# Peer review of "The distribution and isotopomeric characterization of nitrous oxide in the Eastern Gotland Basin (central Baltic Sea)"

_EGUsphere, 2025_

## Community Comment (CC1)

General comment

N2O is one of the most important greenhouse gases, and the ocean plays an important role it the global budget. The production of this gas in the water column will be boosted at low oxygen concentration but consumed when the oxygen concentration drop to certain criteria. Therefore the water with low oxygen concentration is generally important source for N2O. Base on the information provided by the authors. The Baltic Sea is good spot for N2O biogeochemical cycle study in the low oxygen concentration, and the authors use state-of-the-art technique such as isotopemer analysis to reveal the possible N2O production, which will of course promote our understanding of the N2O dynamics in this study region. Unfortunately, I think there are some major issues in this manuscript, and I think the author should make more effort to revise it before it is acceptable for the Journal. One of my biggest concern is that the authors divide the water column in two types of water, oxic and suboxic, and the suboxic water also divided in two groups bases on whether or not there is sulfide detectable. However, I can't not tell is this a reasonable way since the there are very limit information for the dissolve oxygen distribution patterns in the study section, only section with small figure in figure 2a, and a station profile figure 2b. However, from Figure 2 and Figure 3, I suppose that there may be annoxic or oxygen-deficient layer presented in the study region. So, the concentration of oxygen should be clearly displayed, and better categorized, this will provide some very important information for the readers. Secondly, I think the manuscipt lacks figure of hydrographic parameters such as temperature and salinity, since the authors mentioned the hydrographic process such as MBI, and it may provide some important information such as the the authors mentioned in the manusript, like, how the oxygen "intrude" the region. Finally, as a whole, I don't think that the authors give the readers a very clear and solid conclusion, there may be many reasons lead to this situation, including poor preparing of the figure, which in turn lead to incomplete description of phenomenons, and skill of writing, which stop the authors from well describe the phenomenon, and so on. Hence, I think that the authors should carefully reanalysis the dataset, redraw the figures and reorganized

the logic and language of this manuscipt. A major revision is needed before the paper is suitable for publication by the journal.

Specific comment

Line 55, signature of the nitrate or nitrite... I think reference is needed here.

Line 69 5-15nM..., if this is general range, I suggest a wilder range since the polar ocean may have higher N2O concentration

Line 75-81 for the first question, I feel there is inconsistent with last paragraph. In the last paragraph, it seems there are colleauges concluded that nitrification is the predominated N2O source, whereas the authors want to answer the dominant pathways, any new insight we can obtain in this study?

Line 82 this Manuscipt is not a methodology paper, so I think this question is put forward inappropriate here, if the authors is not confident to use the method here, they should carry out a study to estimate of effectiveness of this method in advance. Line 130 Is this means that the sample is bubble free before capping, generally, when butyl stoppers and aluminium crimps was capped, it will easily introduce bubble, so i guess bubble free should be make sure after capping?

Figure 2.    as mentioned before, hydrographic figures should be provided here, and the oxygen profile of each stations should be added to figure 3

Line 174 O2 between 9-2umol? Typo?

Figure 3, the scale of x y axis of each figure should agree with each other, so it is easier for the reader to compare them.

Line 229 "Significant" is a word used only after statistic data analysis performed, it generally should followed with a criteria such as p value

Line 264 I think the first sentence should be rewritten

Line 396 the conclusion should be rewritten, there is no solid conlusion in current format, for example, the sentence "our results demonstrated the spatial variability..." what kind of variability? Moreover, half of the paragraph is about future work, generally, it will only be one or two sentence for future work

---

## Author Comment (AC1)

Reviewer comments are written in bold italics; our answers are kept in plain font.

General comment: This manuscript by Bardhan et al. reports N2O concentration and multi-isotope abundance from the central Baltic Sea. They found that, in oxic waters, N2O accumulates with isotopic signatures indicating ammonia oxidation-derived N2O source; in suboxic or anoxic waters, they found elevated isotopes signatures and attributed the N2O processes (small concentration) to consumption by denitrification or even chemodenitrification. Overall, this publication is easy to follow. However, as its current form, this study does not deliver a strong enough conclusion.

We thank the reviewer for their constructive and helpful comments and suggestions about our paper. Following, we reply to each issue individually, and explain the changes we will make to the revised manuscript to meet the reviewer's criticism.

- With samples during one cruise, the study heavily relies on the fragmented isotope results (some are not available due to low concentration) for discussion of N2O processes. Little information from other parameters are implemented for supporting such explanations, including salinity, temperature and even DOC/DOM contents. In addition, more side-by-side comparison with other N2O isotope studies and summary in figures/tables may be necessary.
  - Thank you for the suggestion. We have added salinity and temperature data under Supplementary Information. We do not have DOC/DOM data. We have now also included a paragraph on N2O isotope studies from some aquatic systems to be included in the revised version:
  - Studies on N2O isotope data are scarce, especially from fresh and brackish water systems. Ho et al. (2023) used a combination of N2O and NO3- isotopic data from the urbanized Scheldt estuary in Europe and observed denitrification to be the dominant pathway of N2O production. Ammonia oxidation, on the other hand, was the most important source of N2O in the eutrophic Pearl River Estuary in China (Zheng et al., 2024). The isotope ratios of N2O identified submarine groundwater discharge to deliver N2O -laden water to the shallow salt-wedge Werribee River estuary in Australia (Wong et al., 2020). Thus N2O isotopic data can shed light on pathways of production, consumption as well as sources of this trace gas.

**References:**

- 1. Ho, L., Barthel, M., Harris, S., Vermuelen, K., Six, J., Bodé, S., Boeckx, P., and Goethals, P.: Unravelling spatiotemporal N2O dynamics in an urbanized estuary system using natural abundance isotopes, Water Research, 247, 120771, 2023.
- 2. Zheng, Y., Zhan, L., Ji, Q., and Ma, X.: Seasonal isotopic and isotopmeric signatures of nitrous oxide produced microbially in a eutrophic estuary, Marine Pollution Bulletin, 204, 116528, 2024.
- 3. Wong, W. W., Lehmann, M. F., Kuhn, T., Frame, C., Poh, S. C., Cartwright, I., and Cook, P. L.: Nitrogen and oxygen isotopomeric constraints on the sources of nitrous oxide and the role of submarine groundwater discharge in a temperate eutrophic saltwedge estuary, Limnology and Oceanography, 66(4), 1068-1082, 2020.

- The discussion was formulated in a simple and thus uncertain way. I am not totally sure hydrogen sulfide detection is enough for identifying specific regions of biogeochemical cycling in the ocean water. Further, the interpretation of the isotope signatures need to be revised: instead of calculating kinetic isotope effects, it is highly important to consider both N2O production and consumption at every station. Even for large production of N2O (net concentration), consumption is always happening.
  - The existing knowledge from Baltic Sea shows chemolithoautotrophic denitrification to be an important process and the presence of H2S is critical for this pathway to take place. Looking at the data from the suboxic depths, we observed a distinct pattern between the study sites with and without detectable sulfide levels. This is why, it made sense to group the data in this fashion.
  - We agree that the simultaneous production and consumption of N2O are quite common and frequently observed in oxygen deficient zones. We also observed the possible production of N2O by fungi and reduction by bacteria in our study site. So we did not calculate kinetic isotope effects, but rather the apparent isotope effects (combining production and consumption) similar to Wenk et al. 2016. We also acknowledge the limitations of the closed-system Rayleigh approach in field studies. We have added this clarification in the revised manuscript.

**Reference:**

- Wenk, C.B., Frame, C. H., Koba, K., Casciotti, K. L., Veronesi, M., Niemann, H., Schubert, C., Yoshida, N., Toyoda, S., Makabe, A., Zopfi, J., and Lehmann, M. F.: Differential N2O dynamics in two oxygen-deficient lake basins revealed by stable isotope and isotopomer distributions, Limnol. Oceanogr., 61, 135-1739, 2016.

**Specific comments:**

Line 12: "\delta values" should not be written as "d".

Thank you. Corrected.

Line 60-64: Strong reduction of  $N_2O$  will also result in enrichment of SP.

Added.

Line 228-230: Keeling plot approach is based on the assumption of simple mixing between source and background, which is clearly not the case here. The production and consumption processes may be quite different from station to station and long the vertical profiles.

We agree. We are using the lack of significant linear relationships in the Keeling plots to stress on the importance of atmospheric N2O in the oxic waters and to show variability among stations.

Line 379-395: Regarding NosZ I and II genes and the likely regulatory mechanism on N2O is beyond the scope of the study and the collect observation evidence. I suggest to leave them out.

The reason we wrote this discussion was to include all possible pathways/players that may have an impact on the isotopic signature of  $N_2O$ . This highlights the need to include these investigations in future studies. However, taking the reviewer's comment into consideration, we will reduce this paragraph to a few sentences and mention that it is beyond the scope of our study.

---

## Author Comment (AC2)

**General comment**

N2O is one of the most important greenhouse gases, and the ocean plays an important role it the global budget. The production of this gas in the water column will be boosted at low oxygen concentration but consumed when the oxygen concentration drop to certain criteria. Therefore the water with low oxygen concentration is generally important source for N2O. Base on the information provided by the authors. The Baltic Sea is good spot for N2O biogeochemical cycle study in the low oxygen concentration, and the authors use state-of-the-art technique such as isotopemer analysis to reveal the possible N2O production, which will of course promote our understanding of the N2O dynamics in this study region. Unfortunately, I think there are some major issues in this manuscript, and I think the author should make more effort to revise it before it is acceptable for the Journal. One of my biggest concern is that the authors divide the water column in two types of water, oxic and suboxic, and the suboxic water also divided in two groups bases on whether or not there is sulfide detectable. However, I can't not tell is this a reasonable way since the there are very limit information for the dissolve oxygen distribution patterns in the study section, only section with small figure in figure 2a, and a station profile figure 2b. However, from Figure 2 and Figure 3, I suppose that there may be annoxic or oxygen-deficient layer presented in the study region. So, the concentration of oxygen should be clearly displayed, and better categorized, this will provide some very important information for the readers.

We thank the reviewer for their constructive and helpful comments and suggestions about our paper. Following, we reply to each issue individually, and explain the changes we will make to the revised manuscript to meet the reviewer's criticism. The existing knowledge from Baltic Sea shows chemolithoautotrophic denitrification to be an important process and the presence of H2S is critical for this pathway to take place. Looking at the data from the suboxic depths, we observed a distinct pattern between the study sites with and without detectable sulfide levels. This is why, it made sense to group the data in this fashion. To make the dissolved oxygen (DO) profiles more prominent for the readers, we have now added a DO plot for each station in Figure 3 of the revised manuscript. The oxygen concentrations are also clearly displayed.

Secondly, I think the manuscript lacks figure of hydrographic parameters such as temperature and salinity, since the authors mentioned the hydrographic process such as MBI, and it may provide some important information such as the the authors mentioned in the manuscript, like, how the oxygen "intrude" the region. Finally, as a whole, I don't think that the authors give the readers a very clear and solid conclusion, there may be many reasons lead to this situation, including poor preparing of the figure, which in turn lead to incomplete description of phenomenons, and skill of writing, which stop the authors from well describe the

phenomenon, and so on. Hence, I think that the authors should carefully reanalysis the dataset, redraw the figures and reorganized the logic and language of this manuscript. A major revision is needed before the paper is suitable for publication by the journal.

We have added temperature and salinity data for all stations under Supporting Information.

The MBI is an important and unique aspect of the Baltic Sea biogeochemistry. It is an importance source of oxygen to the deeper basins. Many studies have documented the MBI in details . We have modified the text in the following manner: "Occasionally, the North Sea waters flow in over the sills and flush the deeper basins. These inflow events are known as MBIs (major Baltic inflow), and they bring oxygen-rich and saline waters to the deeper basins of the southern and central Baltic Sea. In the recent past, the MBIs have been occurring roughly once in a decade (Gräwe et al., 2015) although this statement has been questioned by Mohrholz (2018) who found a decadal variability of MBIs with a timescale of 25-30 years. The most recent MBI before our sampling campaign, which was also the third largest one in 60 years, occurred in December 2014 (Liblik et al., 2018; Dellwig et al., 2021). Walter et al. (2006) studied the N2O dynamics during the MBI event of 2003. These studies have visual representations depicting the flow of the North Sea waters into the deeper basins of the Baltic Sea. Our study was during a stagnant period."

We have modified the Conclusion section. The datasets have been carefully reanalysed, figures have been redrawn and the language has also been restructured.

Specific comment

Line 55, signature of the nitrate or nitrite... I think reference is needed here.

Added.

Bourbonnais, A., Letscher, R., Bange, H., Echevin, V., Larkum, J., Mohn, J., Yoshida, N. and Altabet, M.: N2O production and consumption from stable isotopic and concentration data in the Peruvian coastal upwelling system, Global Biogeochem. Cy., 31(4), 678-698, 2017.

Line 69 5-15nM..., if this is general range, I suggest a wilder range since the polar ocean may have higher N2O concentration

We omitted this part of the sentence.

Line 75-81 for the first question, I feel there is inconsistent with last paragraph. In the last paragraph, it seems there are colleagues concluded that nitrification is the predominated N2O source, whereas the authors want to answer the dominant pathways, any new insight we can obtain in this study?

Walter et al. was published in 2006 and the cruise was conducted immediately preceding a North Sea inflow event. We have mentioned this in Line 81 now. Our data is more reflective of the stagnant conditions that exist for the majority of the time.

Line 82 this Manuscript is not a methodology paper, so I think this question is put forward inappropriate here, if the authors is not confident to use the method here, they should carry out a study to estimate of effectiveness of this method in advance.

We changed the language to "How to interpret N2O pathways using stable isotopic data, including site preference, as analytical tools?"

Line 130 Is this means that the sample is bubble free before capping, generally, when butyl stoppers and aluminium crimps was capped, it will easily introduce bubble, so i guess bubble free should be make sure after capping?

We ensured each vial was bubble free after capping by closing each vial under running water from the CTD.

Figure 2. as mentioned before, hydrographic figures should be provided here, and the oxygen profile of each stations should be added to figure 3

We have added temperature and salinity data of all stations under supplementary information.

Oxygen profiles of each stations have been added to Figure 3.

Line 174 O2 between 9-2umol? Typo?

Yes. Corrected to 2-9 uM

Figure 3, the scale of x y axis of each figure should agree with each other, so it is easier for the reader to compare them.

Corrected.

Line 229 "Significant" is a word used only after statistic data analysis performed, it generally should followed with a criteria such as p value

Removed.

**Line 264 I think the first sentence should be rewritten**

Changed to "..the  $\delta^{15}N_{\text{bulk}}$  and  $\delta^{18}O$  of  $N_2O$  in the oxic surface waters closely resembled those of trophospheric  $N_2O$ "

Line 396 the conclusion should be rewritten, there is no solid conclusion in current format, for example, the sentence "our results demonstrated the spatial variability..." what kind of variability? Moreover, half of the paragraph is about future work, generally, it will only be one or two sentence for future work

We are referring to the variability between Gotland and Bornholm basins that we discussed.

Section 5 has been renamed to "Concluding remarks and future scope" as we think these knowledge gaps are important information.